# SoGCN: Second-Order Graph Convolutional Networks

## Abstract

We introduce a second-order graph convolution (SoGC), a maximally localized kernel, that can express a polynomial spectral filter with arbitrary coefficients. We contrast our SoGC with vanilla GCN, first-order (one-hop) aggregation, and higher-order (multi-hop) aggregation by analyzing graph convolutional layers via generalized filter space. We argue that SoGC is a simple design capable of forming the basic building block of graph convolution, playing the same role as $3 \times 3$ kernels in CNNs. We build purely topological Second-Order Graph Convolutional Networks (SoGCN) and demonstrate that SoGCN consistently achieves state-of-the-art performance on the latest benchmark. Moreover, we introduce the Gated Recurrent Unit (GRU) to spectral GCNs. This explorative attempt further improves our experimental results.

## 1 Introduction

Deep localized convolutional filters have achieved great success in the field of deep learning. In image recognition, the effectiveness of $3 \times 3$ kernels as the basic building block in Convolutional Neural Networks (CNNs) is shown both experimentally and theoretically (Zhou, 2020). We are inspired to search for the maximally localized Graph Convolution (GC) kernel with full expressiveness power for Graph Convolutional Networks (GCNs).

Most existing GCN methods utilize localized GCs based on one-hop aggregation scheme as the basic building block. Extensive works have shown performance limitations of such design due to over-smoothing (Li et al., 2018; Oono & Suzuki, 2019; Cai & Wang, 2020). In vanilla GCNs (Kipf & Welling, 2017) the root cause of its deficiency is the lumping of the graph node self-connection with pairwise neighboring connections. Recent works of Xu et al. (2019); Dehmamy et al. (2019); Ming Chen et al. (2020) disentangle the effect of self-connection by adding an identity mapping (so-called first-order GC). However, its lack of expressive power in filter representation remains (Abu-El-Haija et al., 2019). The work of (Ming Chen et al., 2020) conjectured that the ability to express a polynomial filter with arbitrary coefficients is essential for preventing over-smoothing.

A longer propagation distance in the graph facilitates GCNs to retain its expressive power, as pointed out by (Liao et al., 2019; Luan et al., 2019; Abu-El-Haija et al., 2019). The minimum propagation distance needed to construct our building block of GCN remains the open question. We show that the minimum propagation distance is two: a two-hop graph kernel with the second-order polynomials in adjacency matrices is sufficient. We call our graph kernel Second-Order GC (SoGC).

We introduce a Layer Spanning Space (LSS) framework to quantify the expressive power of multi-layer GCs for modeling a polynomial filter with arbitrary coefficients. By relating low-pass filtering on the graph spectrum (Hoang & Maehara, 2019) with over-smoothing, one can see the lack of filter representation power (Ming Chen et al., 2020) can lead to the performance limitation of GCN.

Using the LSS framework, we show that SoGCs can approximate any linear GCNs in channel-wise filtering. Furthermore, higher-order GCs do not contribute more expressiveness, and vanilla GCN or first-order GCs cannot represent all polynomial filters in general. In this sense, SoGC is the maximally localized graph kernel with the full representation power.

To validate our theory, we build Second-Order Graph Convolutional Networks (SoGCN) using SoGC kernels. Our SoGCN using simple graph topological features consistently achieves state-

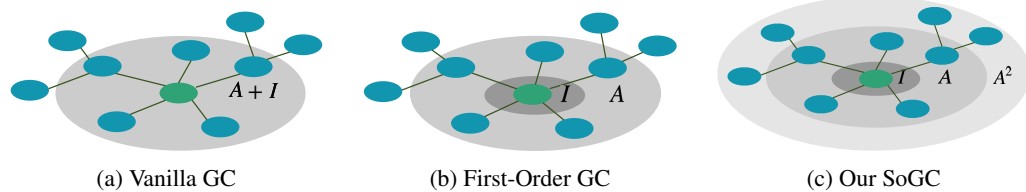

Figure 1: Vertex domain interpretations of vanilla GC, First-Order GC and SoGC. Denote $I$ as the zero-hop aggregator, $A$ the first-hop aggregator and $A^2$ the second-hop aggregator. Nodes in the same colored ring share the same weights. (a) $I$ and $A$ of vanilla GC share the same weights. (b) First-order GC disentangles $I$ from $A$. (c) SoGC introduces new weights for $A^2$ in addition.

of-the-art performance on the GNN benchmark datasets (Dwivedi et al., 2020), including citation networks, super-pixel classification, and molecule regression.

To our best knowledge, this work is the first study that identifies the importance of the two-hop neighborhood in the context of GCNs' ability to express a polynomial filter with arbitrary coefficients. Our model is a special but non-trivial case of Defferrard et al. (2016). Kipf & Welling (2017) conducted an ablation study with GC kernels of different orders but missed the effectiveness of the second-order relationships. The work of Abu-El-Haija et al. (2019) talked about muti-hop graph kernels; however, they did not identify the critical importance of the second-order form. In contrast, we clarify the prominence of SoGCs in theories and experiments.

Our research on graph convolution using pure topologically relationship is orthogonal to those uses geometric relations (Monti et al., 2017; Fey et al., 2018; Pei et al., 2020), or those with expressive edge features (Li et al., 2016; Gilmer et al., 2017; Corso et al., 2020), and hyper-edges (Morris et al., 2019; Maron et al., 2018; 2019). It is also independent with graph sampling procedures (Rong et al., 2019; Hamilton et al., 2017; Li et al., 2019).

## 2 PRELIMINARIES

We begin by reformulating spectral GCNs and introducing our notation. We are interested in a finite graph set $\mathcal{G} = \{G_1, \cdots, G_{|\mathcal{G}|}\}$. Assume each graph $G \in \mathcal{G}$ is simple and undirected, associated with a finite vertex set $\mathcal{V}(G)$, an edge set $\mathcal{E}(G) = \{(u, v) : \forall u \leftrightarrow v\}$, and a symmetric normalized adjacency matrix $\boldsymbol{A}(G)$ (Chung & Graham, 1997; Shi & Malik, 2000). Without loss of generality and for simplicity, $|\mathcal{V}(G)| = N$ for every $G \in \mathcal{G}$. Single-channel features $\boldsymbol{x} \in \mathbb{R}^N$ supported in graph $G \in \mathcal{G}$ is a vectorization of function $\mathcal{V}(G) \to \mathbb{R}$.

Graph Convolutions (GCs) is known as Linear Shift-Invariant (LSI) operators to adjacency matrices (Sandryhaila & Moura, 2013). By this definition, GCs can extract features regardless of where local structures fall. Given parameter space $\Omega \subseteq \mathbb{R}$, we write a single-channel GC (Sandryhaila & Moura, 2013; Defferrard et al., 2016) as a mapping $f_{\boldsymbol{\theta}} : \mathcal{G} \times \mathbb{R}^N \to \mathbb{R}^N$ such that [1]:

$$f_{\boldsymbol{\theta}}(G, \boldsymbol{x}) = \sum_{k=0}^{K} \theta_k \boldsymbol{A}(G)^k \boldsymbol{x}, \tag{1}$$

where $\boldsymbol{\theta} = \begin{bmatrix} \theta_0 & \cdots & \theta_K \end{bmatrix}^T \in \Omega^{K+1}$ parameterizes the GC. $K$ reflects the localization of $f_{\boldsymbol{\theta}}$: a linear combination of features aggregated by $\boldsymbol{A}(G)^k$. Moreover, we reformulate two popular models, vanilla GC (Figure 1a) and first-order GC (Figure 1b), as below:

$$f_0(G, \boldsymbol{x}) = \theta \left( \boldsymbol{A}(G) + \boldsymbol{I} \right) \boldsymbol{x}, \quad f_1(G, \boldsymbol{x}) = \left( \theta_1 \boldsymbol{A}(G) + \theta_0 \boldsymbol{I} \right) \boldsymbol{x}. \tag{2}$$

The general spectral GCNs stack $L$ layers of GCs (Equation 1) with nonlinear activations. Let $f^{(l)}$ be GC layers with parameters $\boldsymbol{\theta}^{(l)} \in \Omega^{K+1}, l \in [L]$, the single-channel GCNs can be written as:

$$F(G, \boldsymbol{x}) = g \circ f^{(L)} \circ \sigma \circ f^{(L-1)} \circ \cdots \circ \sigma \circ f^{(1)}(G, \boldsymbol{x}), \tag{3}$$

---

[1]We can replace the Laplacian matrix $\boldsymbol{L}$ in Defferrard et al. (2016) with the normalized adjacency matrix $\boldsymbol{A}$ since $\boldsymbol{L} = \boldsymbol{I} - \boldsymbol{A}$.

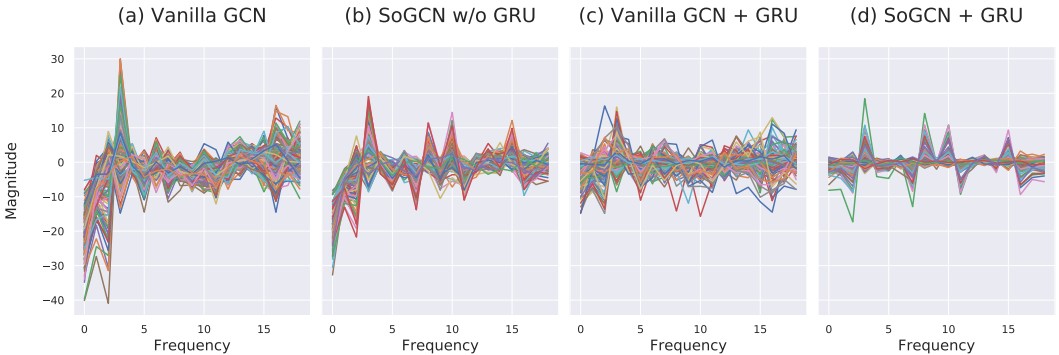

Figure 2: Visualizing output activation in graph spectrum domain for vanilla GCN, SoGCN, and GRU variants. The test is conducted on a graph from the ZINC dataset. The spectrum is defined as a projection of activation functions on the graph eigenvectors. SoGCN preserved higher-order spectrum, while vanilla GCN shows over-smoothing. See Appendix F for more visualizations on the ZINC dataset.

where $\sigma$ is an element-wise activation function, the superscripts denote the corresponding layer number, $g$ is a task-specified readout function (e.g., softmax), the inputs are graph $G \in \mathcal{G}$ and signals $\boldsymbol{x} \in \mathbb{R}^N$. The compositionality principle of deep learning suggests $L$ being large, while $K$ being small and localized (LeCun et al., 2015).

## 3 OVERVIEW: SECOND-ORDER GRAPH CONVOLUTION

We are interested in the overall graph convolution network's representation power of expressing a polynomial filter (Equation 1) with arbitrary coefficients. A multi-layer GCN approximate a $K$-order polynomial filter $\sum_{k=0}^{K} \theta_k \boldsymbol{A}(G)^k, \theta_k \in \Omega, k = 0, \cdots K$, by stacking basic building blocks of graph convolution (GC) layers (Wu et al., 2019; Ming Chen et al., 2020).

We formally define the second-order GC (SoGC) using the second-order polynomials of adjacency matrices:

$$f_2(G, \boldsymbol{x}) = \left(\theta_2 \boldsymbol{A}(G)^2 + \theta_1 \boldsymbol{A}(G) + \theta_0 \boldsymbol{I}\right) \boldsymbol{x}, \tag{4}$$

where $\theta_i \in \mathbb{R}, i = 0, 1, 2$ are trainable paremeters in the context of machine learning. Its vertex-domain interpretation is illustrated in Figure 1c. At first glance, it seems that we could stack two one-hop graph convolution (GC) kernels to approximate a SoGC. However, as shown in Section 4.3, that is not the case.

The critical insight is that graph filter approximation can be viewed as a polynomial factorization problem. It is known that any univariate polynomial can be factorized into sub-polynomials of degree two. Based on this fact, we show by stacking enough SoGCs (and varying their parameters) can achieve decomposition of any polynomial filters.

In contrast, first-order GCs are not universal approximators; two stacked one-hop GCs cannot model every two-hop GC. Polynomial filter completeness of SoGC leads to better performance of GCNs. As shown in Figure 2, networks built with SoGC can overcome over-smoothing and extract features on high-frequency bands. In the next section, we demonstrate our formal arguments on polynomial approximation.

## 4 REPRESENTATION POWER ANALYSIS

### 4.1 LAYER SPANNING SPACE FRAMEWORK

To illustrate the representation power of GC layers, we establish a Layer Spanning Space (LSS) framework to study the graph filter space spanned by stacking multiple graph kernels.

First, we present our mathematical devices in Definition 1, 2 with Lemma 1 as below.

**Definition 1.** *Suppose the parameter space $\Omega = \mathbb{R}$. The Linear Shift-Invariant (LSI) graph filter space of degree $K > 0$ with respect to a finite graph set $\mathcal{G}$ is defined as $\mathcal{A} = \left\{ f_{\boldsymbol{\theta}} : \mathcal{G} \times \mathbb{R}^N \to \mathbb{R}^N, \forall \boldsymbol{\theta} \in \mathbb{R}^{K+1} \right\}$, where $f_{\boldsymbol{\theta}}$ follows the definition in Equation 1.*

**Definition 2.** *Let spectrum set $\mathcal{S}(\mathcal{G}) = \{\lambda : \lambda \in \mathcal{S}(\boldsymbol{A}(G)), \forall G \in \mathcal{G}\}$, where $\mathcal{S}(\boldsymbol{A})$ denotes the eigenvalues of $\boldsymbol{A}$. Define spectrum capacity $\Gamma = |\mathcal{S}(\mathcal{G})|$. In particular, $\Gamma = (N-1)|\mathcal{G}|$ if every graph adjacency matrix has no common eigenvalues other than $1$.*

**Lemma 1.** *$\mathcal{A}$ of degree $K > 0$ has dimension $\min\{K+1, \Gamma\}$ as a vector space.*

Proof of Lemma 1 follows from Theorem 3 of Sandryhaila & Moura (2013). The complete version can be found in Appendix C. Here, we induce a finite-dimension filter space by Lemma 1.

For simplicity, we will model the linear composition of filters to analyze its representation power. The nonlinear activation effects are beyond the scope of this work. Following Definition 1, let $\mathcal{A}$ be the full filter space of degree $\Gamma - 1$ and $\mathcal{B}$ be the low-level filter space as a set of polynomials in adjacency matrices (Equation 1). Denote the GC at $l$-th layer by $f^{(l)} \in \mathcal{B}$, then we yield the LSS of stacking $L$ layers as below:

$$\mathcal{B}^L = \left\{ f : f(G, \boldsymbol{x}) = f^{(L)} \circ \cdots \circ f^{(1)}(G, \boldsymbol{x}) = \prod_{l=1}^{L} p^{(l)}(\boldsymbol{A}(G))\boldsymbol{x} \right\}, \tag{5}$$

where $p^{(l)}(x)$ varies in a certain class of polynomials. We can assess the expressive capability of GC layers by comparing the LSS with $\mathcal{A}$. Kernels in $\mathcal{B}$ have full representation power if $\mathcal{A} \subseteq \mathcal{B}^L$. We are interested in $\mathcal{B}_K$, which denotes all localized filters of degree at most $K$. The LSS of $\mathcal{B}_K$ is modeled as:

$$\mathcal{B}_K^L = \left\{ f : f(G, \boldsymbol{x}) = \prod_{l=1}^{L} \sum_{k=0}^{K} \theta_k^{(l)} \boldsymbol{A}(G)^k \boldsymbol{x}, \theta_k^{(l)} \in \mathbb{R} \right\}, \tag{6}$$

where the number of layers is bounded by $L \leq \lceil (\Gamma - 1)/K \rceil$, according to Lemma 1.

## 4.2 UNIVERSAL REPRESENTATION POWER OF SoGC

In this section, we present Theorem 1 to demonstrate the universal representation power of SoGCs as claimed in Section 3. Formally, we add superscripts to Equation 4 to indicate the layer number. Then we leverage a fundamental polynomial factorization theorem to conclude Theorem 1 as below.

**Theorem 1.** *For any $f \in \mathcal{A}$, there exists $f_2^{(l)} \in \mathcal{B}_2$ with coefficients $\theta_0^{(l)}, \theta_1^{(l)}, \theta_2^{(l)} \in \mathbb{R}, l = 1, \cdots, L$ such that $f = f_2^{(L)} \circ \cdots \circ f_2^{(1)}$ where $L \leq \lceil (\Gamma - 1)/2 \rceil$.*

The complete proof is presented Appendix D. Theorem 1 can be regarded as the universal approximation theorem of linear GCNs, which implies multi-layer SoGCs have full filter expressiveness.

Theorem 1 also demonstrates how GCNs with SoGCs benefit from depth, which coincides with the view of Dehmamy et al. (2019). Figure 3a verifies our SoGCN can overcome over-smoothing and successfully utilize depth to attain performance gain.

## 4.3 REPRESENTATION POWER OF OTHER GRAPH CONVOLUTION

In this section, we show that vanilla and first-order GCs lack expressiveness, while higher-order GCs reduce compactness and increase fitting difficulty.

**Vanilla vs. second-order.** Extensive works have shown the performance deficiency of vanilla GCNs (Hoang & Maehara, 2019; Luan et al., 2019; Oono & Suzuki, 2019; Cai & Wang, 2020). Based on the LSS framework, we can point out a similar issue but from a novel perspective. Let us write $f_0^{(l)}(G, \boldsymbol{x}) = \theta^{(l)}(\boldsymbol{A}(G) + \boldsymbol{I})\boldsymbol{x} \in \mathcal{B}_0$ as the $l$-th GC layer[2]. Then $L$ of them can represent a LSS as follows:

$$\mathcal{B}_0^L = \left\{ f : f(G, \boldsymbol{x}) = \theta \sum_{l=0}^{L} \binom{l}{L} \boldsymbol{A}(G)^l \boldsymbol{x} \right\}, \tag{7}$$

---

[2]Notice that we use $\mathcal{B}_0$ to denote the filter space with lumping of self-connection with pairwise neighbor nodes, since the zero-degree ones are too trivial. We assume the renormalization trick can be merged into $\theta$.

by letting $\theta = \theta^{(L)} \cdots \theta^{(1)}$. No matter how large $L$ is or how a optimizer tunes the parameters $\theta^{(l)}$, $\dim \mathcal{B}_0^L = 1$ which signifies $\mathcal{B}_0^L$ degenerates to a negligible subspace of $\mathcal{A}$.

**First-order vs. second-order.** We denote first-order GCs as $f_1^{(l)}(G, \boldsymbol{x}) = (\theta_1^{(l)} \boldsymbol{A}(G) + \theta_0^{(l)} \boldsymbol{I}) \boldsymbol{x} \in \mathcal{B}_1$. In the spirit of Section 4.1, write the LSS as:

$$\mathcal{B}_1^L = \left\{ f : f(G, \boldsymbol{x}) = \prod_{l=1}^{L} \left( \theta_1^{(l)} \boldsymbol{A}(G) + \theta_0^{(l)} \boldsymbol{I} \right) \boldsymbol{x}, \theta_0^{(l)}, \theta_1^{(l)} \in \mathbb{R} \right\}, \tag{8}$$

which is isomorphic to a polynomial space whose elements split over the real domain. Compared with $\mathcal{B}_0^L$ (Equation 7), $\mathcal{B}_1^L$ represents a much larger subset of $\mathcal{A}$. This highlights the importance of the first-order term or the identity mapping mentioned in (Xu et al., 2019; Dehmamy et al., 2019; Ming Chen et al., 2020).

The limitations also become obvious since not all polynomials can be factorized into first-order polynomials. These polynomials only occupy a small proportion in the ambient polynomial space (Li, 2011), which indicates first-order GCs are not universal approximators in general.

**Higher-order vs. second-order.** GCs of degree $K \geq 2$ are called higher-order GCs. They can model multi-hop GCNs such as Luan et al. (2019); Liao et al. (2019); Abu-El-Haija et al. (2019). Higher-order GCs have equivalent expressive power to SoGCs, since they can be reduced to SoGCs as long as coefficient sparsity can be achieved. But this by-product–an uncertain sparsity of coefficients–is not compatible with gradient-based optimization algorithms. Extensive experiments (Defferrard et al., 2016) have shown the ineffectiveness of learning higher-order kernels, because eigenvalues of graph adjacency matrices diminish when powered. This results in a decreasing numerical rank of $\boldsymbol{A}(G)^k$, which prevent higher-order GCs from aggregating larger-scale information. SoGCs can alleviate this problem by preventing the loss of information due to higher-order powering operation. Finally, higher-order GC lacks nonlinearity. SoGCN can bring a better balance between the expressive power of low-level layers and nonlinearity among them.

## 5 SECOND-ORDER GRAPH CONVOLUTIONAL NETWORKS

In this section, we introduce other building blocks of GCNs and establish our Second-Order Graph Convolutional Networks (SoGCN) following the fashion of deep learning. First, we promote SoGC to the multi-channel version analogous to Kipf & Welling (2017). Then we cascade a feature embedding layer, multiple SoGC layers, and append a readout module. Suppose the multi-channel input is $\boldsymbol{X} \in \mathbb{R}^{N \times D}$ supported in graph $G \in \mathcal{G}$, denote the output of $l$-th layer as $\boldsymbol{X}^{(l)} \in \mathbb{R}^{N \times E}$, the final node-level output as $\boldsymbol{Y} \in \mathbb{R}^{N \times F}$, or graph-level output as $\boldsymbol{Y} \in \mathbb{R}^E$, we formulate our novel deep GCN based on SoGC as follows:

$$\boldsymbol{X}^{(0)} = \rho\left(\boldsymbol{X}; \boldsymbol{\Phi}\right), \tag{9}$$

$$\boldsymbol{X}^{(l+1)} = \sigma\left(\boldsymbol{A}(G)^2 \boldsymbol{X}^{(l)} \boldsymbol{\Theta}_2^{(l+1)} + \boldsymbol{A}(G) \boldsymbol{X}^{(l)} \boldsymbol{\Theta}_1^{(l+1)} + \boldsymbol{X}^{(l)} \boldsymbol{\Theta}_0^{(l+1)}\right), \tag{10}$$

$$\boldsymbol{Y} = \tau\left(\boldsymbol{X}^{(L)}; \boldsymbol{\Psi}\right), \tag{11}$$

where $\boldsymbol{\Theta}_i^{(l)} \in \mathbb{R}^{E \times E}, i = 0, 1, 2$ are trainable weights for linear filters; $\rho : \mathbb{R}^{N \times D} \to \mathbb{R}^{N \times E}$ is an equivariant embedder (Maron et al., 2018) with parameters $\boldsymbol{\Phi}$; $\sigma : \mathbb{R}^{N \times E} \to \mathbb{R}^{N \times E}$ is an activation function. For node-level readout, $\tau : \mathbb{R}^{N \times E} \to \mathbb{R}^{N \times F}$ can be a decoder (with parameters $\boldsymbol{\Psi}$) or a nonlinear activation (e.g., softmax) in place of the prior layer. For graph-level output, $\tau : \mathbb{R}^{N \times E} \to \mathbb{R}^E$ should be an invariant readout function (Maron et al., 2018), e.g., channel-wise sum, mean or max (Hamilton et al., 2017). In practice, we adopt ReLU as nonlinear activation (i.e., $\sigma = \text{ReLU}$), a multi-layer perceptron (MLP) as the embedding function $\rho$, another MLP for node regression readout, and sum (Xu et al., 2019) for graph classification readout.

### 5.1 GATED RECURRENT UNIT

Gated Recurrent Unit (GRU) has been served as a basic building block in message-passing GNN architectures (Li et al., 2016; Gilmer et al., 2017; Corso et al., 2020). In this subsection, we explore its application in spectral GCNs.

According to Cho et al. (2014), GRU can utilize gate mechanism to preserve and forget information. We hypothesize that a GRU can be trained to remove redundant signals and retain lost features on the spectrum. This function can be used to alleviate the oversmoothing problem of vanilla GCNs by maintaining information from previous layer and canceling the dominance of low-frequencies. By the same means, GRU can also relieve the side-effect of ReLU, which is proved to be a special low-pass filter (Oono & Suzuki, 2019; Cai & Wang, 2020). Even though piled-up SoGCs attain full expressiveness, we show by our experiment that GRU can still facilitate SoGCN in avoiding noises and enhancing features on the spectrum (Figure 2)

Similar to Li et al. (2016); Gilmer et al. (2017), we appends a shared GRU module after each GC layer, which takes the signal before the GC layer as the hidden state, after the GC layer as the current input. We note that GRU can cooperate with any spectral GCs (Equation 1). When integrated with SoGCN, we call this special variant SoGCN-GRU. We formulate its implementation by replacing Equation 10 with Equation 12 as below.

$$
\begin{aligned}
\boldsymbol{X}_{conv}^{(l+1)} &= \boldsymbol{A}(G)^2 \boldsymbol{X}^{(l)} \boldsymbol{\Theta}_2^{(l+1)} + \boldsymbol{A}(G) \boldsymbol{X}^{(l)} \boldsymbol{\Theta}_1^{(l+1)} + \boldsymbol{X}^{(l)} \boldsymbol{\Theta}_0^{(l+1)}, \\
\boldsymbol{X}^{(l+1)} &= \mathrm{GRU}\left(\mathrm{ReLU}\left(\boldsymbol{X}_{conv}^{(l+1)}\right), \boldsymbol{X}^{(l)}; \boldsymbol{\Omega}\right),
\end{aligned}
\tag{12}
$$

where $\boldsymbol{X}_{conv}^{(l+1)}$ is the input, $\boldsymbol{X}^{(l)}$ represents the hidden state, $\boldsymbol{\Omega}$ denotes parameters of the GRU.

Figure 2 and Figure 6 verify our conjecture. We observe more steady low-frequency component on the spectrum head and more characteristic bands on the high-frequency tail. Our empirical study in Table 3 also indicates the effectiveness of GRU for spectral GCNs in general. Hence, we suggest including this recurrent module as another basic building block of our SoGCNs.

## 5.2 Comparison to Related Work

**Spectral GCNs.** Spectral GCN leverages polynomials in the adjacency matrix to represent graph convolutional layers (Bruna et al., 2014). Many works have been discussing how to design the polynomial and choose its degree to compose a localized GC layer. ChebyNet (Defferrard et al., 2016) approximates graph filters using Chebyshev polynomials. Vanilla GCN (Kipf & Welling, 2017; Wu et al., 2019) further reduces the GC layer to a degree-one polynomial with first-order and constant terms merged. However, these simplifications cause over-smoothing and performance loss. Our SoGC incorporates only one hop longer but obtains the full representation power. This design keeps each layer localized, simple, and easy to implement but makes the whole GCN much more powerful. Our work reveals the critical degree of polynomial filters where kernel size is minimized while filter representation power is maximized.

**Multi-Hop GCNs.** To exploit multi-scale information, Luan et al. (2019) devises Snowball GCN and Truncated Krylov GCN to capture neighborhoods at different distances. To simulate hop delta functions, Abu-El-Haija et al. (2019) repeat mixing multi-hop features to identify more topological information. These models exhibit the strength of multi-hop GCNs over one-hop GCNs while leaving the propagation length as a hyper-parameter. Modeling those multi-hop GCNs as our higher-order models, SoGCN possesses the identical representation power but has fixed size and better localization, making SoGC more suitable to be the basic building block in GCNs. It is noteworthy that, although Abu-El-Haija et al. (2019) investigates the two-hop delta function (a Gabor-like filter), their final proposed solution is only a generic class of multi-hop GCNs. The discussion on two-hop delta functions cannot attain our theoretical results.

**Expressiveness of GCNs.** Most of the works on GCN's expressiveness are restricted to the over-smoothing problem: Li et al. (2018) first poses the over-smoothing problem; Hoang & Maehara (2019) indicates GCNs are no more than low-pass filters; Luan et al. (2019); Oono & Suzuki (2019) demonstrate the asymptotic behavior of feature activation to a subspace; Cai & Wang (2020) examines the decreasing Dirichlet energy. Unlike them, our established LSS framework can identify specific issues of GCNs with algebraic and geometric interpretations. The over-smoothing problem can be formulated as one of our sub-problems (Section 4.3). In this sense, SoGCN solves a more general expressiveness issue than those relieving over-smoothing problem only (Xu et al., 2018; Rong et al., 2019; Chen et al., 2020). Ming Chen et al. (2020) introduces identity and initial mapping to recover filter expressiveness. Their analytic framework is also similar to ours. But we

generalize their filter space to a graph set, and upper bound its dimension. In the meanwhile, our SoGCN's architecture is more lightweight. We investigate the overall expressive power of GCNs by discussing filter completeness. This direction is orthogonal to those studying message-passing GNNs (Scarselli et al., 2008) and Weisfeiler-Leman GNNs (Xu et al., 2019; Morris et al., 2019).

## 6 EXPERIMENTS

Experiments are conducted on the synthetic dataset in Section 6.1 and on the GNN benchmarks (Dwivedi et al., 2020) in Section 6.2.

### 6.1 SYNTHETIC GRAPH SPECTRUM DATASET FOR NODE REGRESSION

To validate the expressiveness of SoGCN, and its power to preserve higher-order graph signal, we build a Synthetic Graph Spectrum (SGS) dataset for the node signal filtering regression task. We construct SGS dataset with random graphs. The learning task is to mimic three types of hand-crafted filtering functions: high-pass, low-pass, and band-pass on the graph spectral space (defined over the graph eigenvectors). There are 1k training graphs, 1k validation graphs, and 2k testing graphs for each filtering function. Each graph is undirected and comprises 80 ˜ 120 nodes. Appendix E covers more details of our SGS dataset. We choose Mean Absolute Error (MAE) as the evaluation metric.

**Experimental Setup.** We compare SoGCN with vanilla GCN (Kipf & Welling, 2017), first-order GCN, and higher-order GCNs on the synthetic dataset.

To evaluate each model's expressiveness purely on the graph kernel design, we remove ReLU activations for all tested models. We adopt the Adam optimizer (Kingma & Ba, 2015) in our training process, with a batch size of 128. The learning rate begins with 0.01 and decays by half once the validation loss stagnates for more than 10 training epochs.

Table 1: The performance of graph node signal regression with *High-Pass*, *Low-Pass*, and *Band-Pass* filters (over graph spectral space) as learning target. Each model has 16 GC layers and 16 channels of hidden layers.

| Model | #Param | Test MAE $\pm$ s.d. | | |
|---|---|---|---|---|
| | | **High-Pass** | **Low-Pass** | **Band-Pass** |
| Vanilla GCN | 4611 | 0.308$\pm$0.006 | 0.317$\pm$0.011 | 0.559$\pm$0.071 |
| Vanilla GCN + ReLU [3] | 4611 | 0.466$\pm$0.002 | 0.457$\pm$0.002 | 0.299$\pm$0.000 |
| 1st-Order GCN | 8467 | 0.036$\pm$0.004 | 0.032$\pm$0.002 | 0.115$\pm$0.008 |
| 3rd-Order GCN | 16179 | 0.021$\pm$0.003 | 0.022$\pm$0.001 | 0.045$\pm$0.008 |
| 4th-Order GCN | 20035 | 0.021$\pm$0.003 | 0.022$\pm$0.002 | 0.049$\pm$0.006 |
| SoGCN | 12323 | 0.021$\pm$0.003 | 0.023$\pm$0.002 | 0.050$\pm$0.004 |

**Results and Discussion.** Table 1 summarizes the quantitative comparisons. SoGCN achieves the superior performance on all of the 3 tasks outperforming vanilla GCN and 1st-order GCN, which implies that SoGC graph convolutional kernel does benefit from explicit disentangling of the $\theta_0 I$ (zero-hop) and $\theta_2 A^2$ (second-hop) terms. Our results also show that higher-order (third-order and fourth-order) GCNs do not improve the performance further, even though they have many more parameters. SoGCN possesses a more expressive representation ability and a good trade-off between performance and model size.

Figure 3 plots MAE results as we vary the depth and channel width of GC layers. Vanilla GCN can benefit from neither depth nor width. First-order GC, SoGC, and higher-order GC can leverage depth to span larger LSS. Figure 3a illustrates the corresponding performance for each graph kernel types. SoGC and higher-order GC both outperform first-order GC as depth increases. Figure 3b shows the

---

[3] This experimental group shows that ReLUs are not always such beneficial on our synthetic dataset.

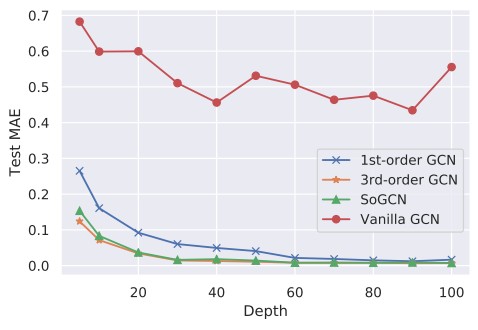

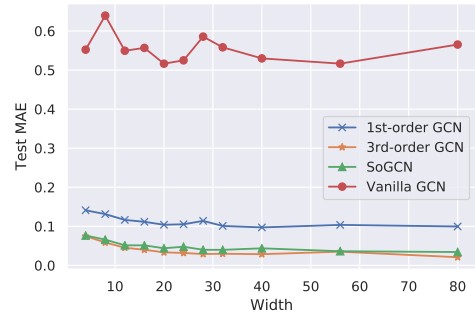

(a) Relations between test MAE and depth.

(b) Relations between test MAE and width.

Figure 3: Relations between test MAE and depth or width. Experiments of both (a) and (b) are conducted on synthetic *Band-Pass* dataset. Each model in (a) has 16 channels per hidden layer with varying depth. Each model in (b) consists of 16 GC layers with varying width.

benefits of SoGC remain as we move to multi-channel construction. Comparing Figure 3a and 3b, we find that depth has larger effect on GCNs.

## 6.2 GNN BENCHMARKS

We follow the benchmarks outlined in Dwivedi et al. (2020) for evaluating GNNs on several datasets across a variety of artificial and real-world tasks. We choose to evaluate our SoGCN on a real-world chemistry dataset (ZINC molecules) for the graph regression task, two semi-artificial computer vision datasets (CIFAR10 and MNIST superpixels) for the graph classification task, and two artificial social network datasets (CLUSTER and PATTERN) for the node classification task.

Table 2: Results and comparison with other GNN models on ZINC, CIFAR10, MNIST, CLUSTER and PATTERN datasets. For ZINC dataset, the parameter budget is set to 500k. For CIFAR10, MNIST, CLUSTER and PATTERN datasets, the parameter budget is set to 100k. **Red**: the best model, **Green**: good models.

| Model | Test MAE ± s.d. | Test ACC ± s.d. (%) | | | |
|---|---|---|---|---|---|
| | ZINC | MNIST | CIFAR10 | CLUSTER | PATTERN |
| Vanilla GCN | 0.367±0.011 | 90.705±0.218 | 55.710±0.381 | 53.445±2.029 | 63.880±0.074 |
| Vanilla GCN-GRU | 0.295±0.005 | 96.020±0.090 | 61.332±0.849 | 57.932±0.168 | 70.194±0.216 |
| GAT | 0.384±0.007 | 95.535±0.205 | 64.223±0.455 | 57.732±0.323 | 75.824±1.823 |
| MoNet | 0.292±0.006 | 90.805±0.032 | 65.911±2.515 | 58.064±0.131 | 85.482±0.037 |
| GraphSage | 0.398±0.002 | **97.312±0.097** | 65.767±0.308 | 50.454±0.145 | 50.516±0.001 |
| GIN | 0.387±0.015 | 96.485±0.252 | 55.255±1.527 | 58.384±0.236 | **85.590±0.011** |
| GatedGCN | 0.350±0.020 | **97.340±0.143** | **67.312±0.311** | 60.404±0.419 | 84.480±0.122 |
| 3WLGNN | 0.407±0.028 [4] | 95.075±0.961 | 59.175±1.593 | 57.130±6.539 | **85.661±0.353** |
| SoGCN | **0.238±0.017** | **96.785±0.113** | **66.338±0.155** | **68.167±1.164** | **85.735±0.037** |
| SoGCN-GRU | **0.201±0.006** | **97.729±0.159** | **68.208±0.271** | **67.994±2.619** | **85.711±0.047** |

**Experimental Setup.** We compare our proposed SoGCN and SoGCN-GRU with state-of-the-art GNNs: vanilla GCN (Kipf & Welling, 2017), GAT (Veličković et al., 2018), MoNet (Monti et al., 2017), GIN (Xu et al., 2019), GraphSage (Hamilton et al., 2017), GatedGCN (Bresson & Laurent, 2017) and 3WL-GNN (Maron et al., 2019). To ensure fair comparisons, we follow the same training and evaluation pipelines (including optimizer settings) and data splits of benchmarks. Furthermore,

---

[4] This is the result of 3WLGNN with 100k parameters. The test MAE of 3WLGNN with 500k parameters is increased to 0.427±0.011.

Table 3: Results of ablation study on ZINC, MNIST and CIFAR10 datasets. Vanilla GCN is the comparison baseline and the number in the ($\uparrow$ ·) and ($\downarrow$ ·) represents the performance gain compared with the baseline.

| Model | Test MAE ± s.d. | Test ACC ± s.d. (%) | |
|---|---|---|---|
| | ZINC | MNIST | CIFAR10 |
| Vanilla GCN | 0.367 ± 0.011 (Baseline) | 90.705 ± 0.218 (Baseline) | 55.710 ± 0.381 (Baseline) |
| 1st-Order GCN | 0.253 ± 0.012 ($\downarrow$ 0.113) | 96.407 ± 0.089 ($\uparrow$ 5.701) | 64.993 ± 0.092 ($\uparrow$ 9.283) |
| SoGCN | 0.238 ± 0.017 ($\downarrow$ **0.129**) | 96.785 ± 0.113 ($\uparrow$ **6.080**) | 66.338 ± 0.155 ($\uparrow$ **10.628**) |
| 3rd-Order GCN | 0.242 ± 0.005 ($\downarrow$ 0.125) | 96.367 ± 0.227 ($\uparrow$ 5.662) | 64.267 ± 0.182 ($\uparrow$ 8.557) |
| 4th-Order GCN | 0.243 ± 0.009 ($\downarrow$ 0.124) | 96.167 ± 0.198 ($\uparrow$ 5.462) | 64.230 ± 0.212 ($\uparrow$ 8.520) |
| Vanilla GCN + GRU | 0.295 ± 0.005 ($\downarrow$ 0.072) | 96.020 ± 0.090 ($\uparrow$ 5.315) | 61.332 ± 0.381 ($\uparrow$ 5.622) |
| 1st-Order GCN + GRU | 0.226 ± 0.015 ($\downarrow$ 0.141) | 96.945 ± 0.093 ($\uparrow$ 6.240) | 62.372 ± 0.522 ($\uparrow$ 6.662) |
| SoGCN + GRU | 0.201 ± 0.006 ($\downarrow$ **0.166**) | 97.729 ± 0.159 ($\uparrow$ **7.024**) | 68.208 ± 0.271 ($\uparrow$ **12.498**) |
| 3rd-Order GCN + GRU | 0.203 ± 0.001 ($\downarrow$ 0.164) | 97.375 ± 0.052 ($\uparrow$ 6.670) | 64.242 ± 0.511 ($\uparrow$ 8.532) |
| 4th-Order GCN + GRU | 0.204 ± 0.004 ($\downarrow$ 0.163) | 97.304 ± 0.296 ($\uparrow$ 6.599) | 64.697 ± 0.341 ($\uparrow$ 8.987) |

we adjust our model's depth and width to ensure it satisfies parameter budgets as specified in the benchmark. Note that we do not use any geometrical information to encode rich graph edge relationship, as in models such as GatedGCN-E-PE. We only employ graph connectivity information for all tested models.

**Results and Discussion.**    Table 2 reports the benchmark results. Our model SoGCN makes small computational changes to GCN by adopting second-hop and zero-hop neighborhoods, and it outperforms models with complex message-passing mechanisms. With GRU module, SoGCN-GRU tops almost all state-of-the-art GNNs on the ZINC, MNIST and CIFAR10 datasets. Whereas, GRU does not lift performance on the CLUSTER and PATTERN datasets for node classification task. As suggested by Li et al. (2018), graph node classification benefits from low-frequency features. That GRU suppresses low-frequency band will result in a slight performance drop on the CLUSTER and PATTERN datasets.

**Ablation Study on High-Order GCNs.**    To contrast the performance gain produced by different orders on the benchmarks, we evaluate 1st-Order GCN, 3rd-Order GCN, 4th-Order GCN as well as their GRU variants on the ZINC, MNIST and CIFAR10 datasets. Table 3 presents the results of our ablation study, which are consistent to our experiments on the synthetic datasets (Section 6.1). As shown by our ablation study, aggregating zero-hop features brings about significant improvements (Vanilla GCN vs. 1st-Order GCN), and adopting the second-hop features further promotes the performance (1st-Order GCN vs. SoGCN). However, high-order GCNs are not capable of boosting the performance over SoGCN. On the contrary, high-order GCs can even lead to the performance decline (3rd-Order GCN vs. 4th-Order GCN vs. SoGCN). On the ZINC and MNIST datasets, we testify GRU's effectiveness for each tested model, but the gain brought by GRU is not as significant as aggregating the second-hop features. On the CIFAR10 dataset, GRU fails to improve performance for 1st-Order GCN and 3rd-Order GCN.

## 7    CONCLUSION

What should be the basic building blocks for GCNs? To answer this, we seek the most localized graph convolution kernel (GC) with full expressiveness. We generalize the filter space to a finite graph set and establish our LSS framework to assess GC layers functioning on different hops. We show the second-order localized graph convolutional filter, called SoGC, possesses the full representation power than one-hop GCs. Thus, it becomes the most localized GC that we adopt as the basic building block to establish our SoGCN. Both synthetic and benchmark experiments exhibit the prominence of our theoretic design. We also make an empirical study on the GRU module cooperating with spectral GCNs. Interesting directions for future work include analyzing two-hop aggregation schemes with message-passing GNNs and proving the universality of nonlinear GCNs.

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

## A    REMARK ON DEFINITION 1

Let us rewrite the $\mathcal{A}$ of degree $K$ in Definition 1:

$$\mathcal{A} = \left\{ f : f(G, \boldsymbol{x}) = \sum_{k=0}^{K} \theta_k \boldsymbol{A}(G)^k \boldsymbol{x}, \forall \theta_k \in \mathbb{R} \right\}, \tag{13}$$

which contains all LSI functions $f : \mathcal{G} \times \mathbb{R}^N \to \mathbb{R}^N$ with adjacency matrix as the graph shift. We show this in the following way. First, all linear transformations $\boldsymbol{H}$ invariant to the shift $\boldsymbol{S}$ should have $\boldsymbol{HS} = \boldsymbol{SH}$. Second, specifying arbitrary graph $G \in \mathcal{G}$, any filter associated with it can be written as below:

$$\boldsymbol{H}(G) = \sum_{k=0}^{K} \theta_k \boldsymbol{A}(G)^k = \boldsymbol{U} \left( \sum_{k=0}^{K} \theta_k \boldsymbol{\Lambda}^k \right) \boldsymbol{U}^T, \tag{14}$$

where $\boldsymbol{A}(G) = \boldsymbol{U}\boldsymbol{\Lambda}\boldsymbol{U}^T$ is the eigendecomposition of $\boldsymbol{A}(G)$. Then we can conclude the shift-invariance property by the following Lemma 2.

**Lemma 2.** *Diagonalizable matrices $\boldsymbol{A}_1$ and $\boldsymbol{A}_2$ are simultaneously diagonalized if and only if $\boldsymbol{A}_1\boldsymbol{A}_2 = \boldsymbol{A}_2\boldsymbol{A}_1$.*

## B    RING ISOMORPHISM: $\mathcal{A} \to \mathcal{T}$

We derive an equivalent form of $\mathcal{A}$, namely construct a tractable space for $\mathcal{A}$. Notice that, this construction is essential to the proof of Lemma 1 and Theorem 1.

Since $\mathcal{G}$ is finite, we can construct a block diagonal matrix $\boldsymbol{T} \in \mathbb{R}^{N|\mathcal{G}| \times N|\mathcal{G}|}$, consisting of all adjacency matrices on the diagonal.

$$\boldsymbol{T} = \begin{bmatrix} \boldsymbol{A}(G_1) & & \\ & \ddots & \\ & & \boldsymbol{A}(G_{|\mathcal{G}|}) \end{bmatrix} \in \mathbb{R}^{N|\mathcal{G}| \times N|\mathcal{G}|}. \tag{15}$$

Hereby, we stop to explain the big picture of Definition 2 with Equation 15. Obviously, the spectrum capacity $\Gamma$ represents the number of eigenvalues of $\boldsymbol{T}$ without multiplicity. Note that, eigenvalues of adjacency matrices signify graph similarity. The spectrum capacity $\Gamma$ identifies a set of graphs by enumerating the structural patterns. Even if the graph set goes extremely large (to guarantee the generalization capability), the distribution of spectrum provide the upper bound of $\Gamma$, so our theories will not lose generality.

Now we get back to construct a matrix space $\mathcal{T}$ from $\mathcal{A}$ via a ring homomorphism $\pi : \mathcal{A} \to \mathcal{T}$:

$$\pi : \sum_{k=0}^{K} \theta_k \boldsymbol{A}(G)^k \mapsto \sum_{k=0}^{K} \theta_k \boldsymbol{T}^k. \tag{16}$$

Recall that a ring homomorphism preserves the "summation" and "multiplication". Concretely, we write the matrix space $\mathcal{T}$ as follows:

$$\mathcal{T} = \left\{ \boldsymbol{H} : \boldsymbol{H} = \sum_{k=0}^{K} \theta_k \boldsymbol{T}^k, \forall \theta_k \in \mathbb{R} \right\}. \tag{17}$$

In the following part, we prove that $\pi$ is an isomorphism.

*Proof.* First, the definition of $\mathcal{T}$ (Equation 17) basically conclude the surjectivity. Second, for any $f_1 \neq f_2 \in \mathcal{A}$ with parameter $\alpha_k, \beta_k \in \mathbb{R}$ where $k = 0, \cdots, K$, there exists $G_j \in \mathcal{G}, \boldsymbol{x} \in \mathbb{R}^N$ such that $f_1(G_j, \boldsymbol{x}) \neq f_2(G_j, \boldsymbol{x})$. And we have their images $\boldsymbol{H}_1 = \pi(f_1), \boldsymbol{H}_2 = \pi(f_2)$. By padding $\boldsymbol{x}$ with zeros like:

$$\boldsymbol{x}' = \begin{bmatrix} \boldsymbol{0}_{N(j-1)}^T & \boldsymbol{x}^T & \boldsymbol{0}_{N(|\mathcal{G}|-j)}^T \end{bmatrix}^T, \tag{18}$$

where $\mathbf{0}_N$ denote the all-zero vector of length $N$. We apply $\boldsymbol{H}_1, \boldsymbol{H}_2$ to $\boldsymbol{x}'$:

$$\boldsymbol{H}_1 \boldsymbol{x}' = \left[\mathbf{0}_{N(j-1)}^T \quad \left(\sum_{k=0}^{K} \alpha_k \boldsymbol{A}(G_j)^k \boldsymbol{x}\right)^T \quad \mathbf{0}_{N(|\mathcal{G}|-j)}^T\right]^T = \left[\mathbf{0}_{N(j-1)}^T \quad f_1(G_j, \boldsymbol{x})^T \quad \mathbf{0}_{N(|\mathcal{G}|-j)}^T\right]^T,$$
(19)

$$\boldsymbol{H}_2 \boldsymbol{x}' = \left[\mathbf{0}_{N(j-1)}^T \quad \left(\sum_{k=0}^{K} \beta_k \boldsymbol{A}(G_j)^k \boldsymbol{x}\right)^T \quad \mathbf{0}_{N(|\mathcal{G}|-j)}^T\right]^T = \left[\mathbf{0}_{N(j-1)}^T \quad f_2(G_j, \boldsymbol{x})^T \quad \mathbf{0}_{N(|\mathcal{G}|-j)}^T\right]^T.$$
(20)

Hence, $\boldsymbol{H}_1 \neq \boldsymbol{H}_2$ concludes the injectivity. $\square$

## C    PROOF OF LEMMA 1

We first show $\mathcal{A}$ is a vector space, then we leverage the isomorphism to have equality $\dim \mathcal{A} = \dim \mathcal{T}$. Figuring out the dimension of $\mathcal{T}$ is much more tractable.

*Proof.* By verifying the linear combination is closed or simply implied from the ring isomorphism $\pi$, $\mathcal{A}$ is at least a vector space

Then Lemma 1 follows from the Theorem 3 of Sandryhaila & Moura (2013). We briefly conclude the proof in the following way. Let $m(x)$ denote the minimal polynomial of $\boldsymbol{T}$. We have $\Gamma = \deg m(x)$. Due to the isomorphism, $\dim \mathcal{A} = \dim \mathcal{T}$.

Suppose $K + 1 < \Gamma$. First, $\dim \mathcal{T}$ cannot be larger than $K + 1$, because $\{\boldsymbol{I}, \boldsymbol{T}, \cdots, \boldsymbol{T}^K\}$ is a spanning set. If $\dim \mathcal{T} < K + 1$, then there exists some polynomial $p(x)$ with $\deg p(x) \leq K$, such that $p(\boldsymbol{A}) = \mathbf{0}$. This contradicts the minimality of $m(x)$. $\dim \mathcal{T}$ can only be $K + 1$.

Suppose $K + 1 \geq \Gamma$. For any $\boldsymbol{H} = h(\boldsymbol{T})$ for some polynomial $h(x)$ with $\deg h(x) \leq K$. By polynomial division, there exists unique polynomials $q(x)$ and $r(x)$ such that

$$h(x) = q(x)m(x) + r(x),$$
(21)

where $\deg r(x) < \deg m(x) = \Gamma$. Insert $\boldsymbol{T}$ into Equation 21:

$$h(\boldsymbol{T}) = q(\boldsymbol{T})m(\boldsymbol{T}) + r(\boldsymbol{T}) = q(\boldsymbol{T})\mathbf{0} + r(\boldsymbol{T}) = r(\boldsymbol{T}).$$
(22)

Therefore, $\{\boldsymbol{I}, \boldsymbol{T}, \cdots, \boldsymbol{T}^{\Gamma-1}\}$ form a basis of $\mathcal{T}$, i.e., $\dim \mathcal{T} = \Gamma$. $\square$

Remark that, we assume each graph contains the same number of vertices only for the sake of simplicity. Lemma 1 still holds when the vertex numbers are varying, since the construction of Equation 15 is independent of this assumption. However, we need the graph set to be finite, otherwise $\Gamma$ might be uncountable. We leave the discussion on infinite graph sets for future study.

## D    PROOF OF THEOREM 1

First, we borrow the concept of $\mathcal{T}$ (Equation 17) in place of $\mathcal{A}$. Then we leverage the following basic yet powerful Lemma 3 to conclude the proof of Theorem 1 straightforwardly.

**Lemma 3.** *Over the field of reals, the degree of an irreducible non-trivial univariate polynomial is either one or two.*

*Proof.* For any $f \in \mathcal{A}$, let us map it to $\boldsymbol{H} = h(\boldsymbol{A})$ through the isomorphism $\pi$ (Equation 16) for some polynomial $h(x)$ with $\deg h(x) \leq \Gamma - 1$ (By Lemma 1).

By Lemma 3, factorize $h(x)$ into series of polynomials with the degree at most two, and then merge first-order polynomials into second-order ones until no paired first-order polynomials remaining. Finally, we obtain the following equation:

$$h(x) = \prod_{l=1}^{\lceil D/2 \rceil} h^{(l)}(x),$$
(23)

where $D = \deg h(x)$. If $D$ is even, $\deg h^{(l)} = 2$ for $l = 1, \cdots, \lceil D/2 \rceil$. Otherwise, there exists some $j \in [L]$ such that $\deg h^{(j)}(x) = 1$. Notice that, the remaining first-order polynomials is at most 1, which indicates the sparsity of coefficients is very low.

Now we obtain filters $\boldsymbol{H}^{(l)} = h^{(l)}(\boldsymbol{T})$ for $l = 1, \cdots, \lceil D/2 \rceil$. The last step is applying the inverse of isomorphism $\pi^{-1}$ to map $\boldsymbol{H}^{(l)} \in \mathcal{T}$ back to $f^{(l)} \in \mathcal{A}$ as below:

$$\pi^{-1} : \sum_{k=0}^{K} \theta_k \boldsymbol{T}^k \mapsto \sum_{k=0}^{K} \theta_k \boldsymbol{A}(G)^k. \tag{24}$$

Recalling Section 4.1, $f^{(l)} \in \mathcal{B}_2$ for $l = 1, \cdots, \lceil D/2 \rceil$. Since $\pi^{-1}$ is also a ring isomorphism, $\boldsymbol{H} = \boldsymbol{H}^{(\lceil D/2 \rceil)} \circ \cdots \circ \boldsymbol{H}^{(1)}$ implies $f = f^{(\lceil D/2 \rceil)} \circ \cdots \circ f^{(1)}$. $\qquad\qquad \square$

Remark that Theorem 1 can be considered as the GCN-version of Theorem 3 in Zhou (2020). It plays a key step in proving the universality of nonlinear CNNs. Our Theorem 1, thus, provides a strong tool for analyzing nonlinear GCNs.

## E    SYNTHETIC GRAPH SPECTRUM (SGS) DATASET

Our SGS dataset works for node signal filtering regression tasks. We designed 3 types of graph signal filters: high-pass (HP), low-pass (LP) and band-pass (BP) filters, as given in Equation 25. For each type, we generate 1000, 1000 and 2000 undirected graphs with graph signals and groundtruth response in the training set, validation set and test set, repectively. Each graph approximately has 80 ˜120 nodes and 80 ˜350 edges. Models trained on each sub-dataset are expected to learn the corresponding filter by supervising the MAE loss.

$$
\begin{aligned}
F_{HP}^*(x) &= \frac{1}{1 + \exp\{-50(x-1)\}} \\
F_{LP}^*(x) &= 1 - \frac{1}{1 + \exp\{-50(x-1)\}} \\
F_{BP}^*(x) &= \frac{-1}{1 + \exp\{-100(x-1.05)\}} + \frac{1}{1 + \exp\{-100(x-0.95)\}}
\end{aligned}
\tag{25}
$$

Undirected graphs are randomly sampled through rejection sampling of edges from complete graphs. In detail, we randomly draw an integer $N$ from $[80, 120]$ as the number of nodes, and then generate a $N \times N$ random matrix $\boldsymbol{B}$ with each element independently sampled from $\mathrm{Unif}(0, 1)$. Set a threshold $\epsilon$ and we can construct an adjacency matrix $\boldsymbol{A}$ by letting $a_{i,j} = 1$ if $b_{i,j} > \epsilon$, otherwise $a_{i,j} = 0$, where $a_{i,j}$ is the element located at the i-th row and j-th column of $\boldsymbol{A}$.

Next, we need to generate spectral signals $\boldsymbol{s}$ for the graph. Independent sampling for each spectrum from a probabilistic distribution will only generate noises. Hence, we synthesize spectrum by summing random functions. We notice that pdf of the beta distribution $\mathrm{Beta}(a, b)$ is a powerful tool to construct diverse curves by tuning shape parameters $a$ and $b$. Also, Gaussian function $\mathrm{Norm}(\mu, \sigma)$ is able to yield diverse bell-shaped curves by tuning $\mu$ and $\sigma$. We sum 2 discretized beta functions, 4 discretized Gaussian functions with random parameters to generate spectral signals. Equation 26 elaborates the generation process and hyper-parameter choosing in our experiments, where $g[x; a, b]$ is the pdf of $\mathrm{Beta}(a, b)$ distribution, $f[x; \mu, \sigma]$ is the pdf of $\mathrm{Norm}(\mu, \sigma)$ distribution.

$$
\begin{aligned}
\boldsymbol{s}[x] &= \sum_{i=1}^{2} g[x/N; a_i, b_i] + \sum_{j=1}^{4} c_j f[x; \mu_j, \sigma_j] \quad x \in [N] \\
a_i, b_i &\sim \mathrm{Unif}\{0.1, 5\} \quad \mu_j \sim \mathrm{Unif}\{0, N\} \quad \sigma_j \sim \mathrm{Unif}\left\{\frac{N}{(j+1)}, \frac{N}{j}\right\}/9 \\
c_j &\sim \frac{\mathrm{Unif}\{0.5, 2\}}{\max_{x \in [N]} f[x; \mu_j, \sigma_j]}
\end{aligned}
\tag{26}
$$

In most real-world cases, graph signals are usually represented in vertex-domain. With a generated graph and its spectral signals, we can retrieve the vertex-domain signals by inverse graph Fourier transformation: perform eigen-decomposition on the normalized adjacency matrix $\tilde{A}$ of the graph to retrieve its graph Fourier basis $\tilde{U}$, then we can find vertex-domain signals by $\tilde{U}s$.

For supervising purpose, we retrieve the groundtruth of each filter's response by applying each filter to the generated spectral signals, namely $F_k^*(s), k \in \{HP, LP, BP\}$. Figure 4 illustrates an example of the generated spectral signals and its groundtruth filtering response of the 3 filters.

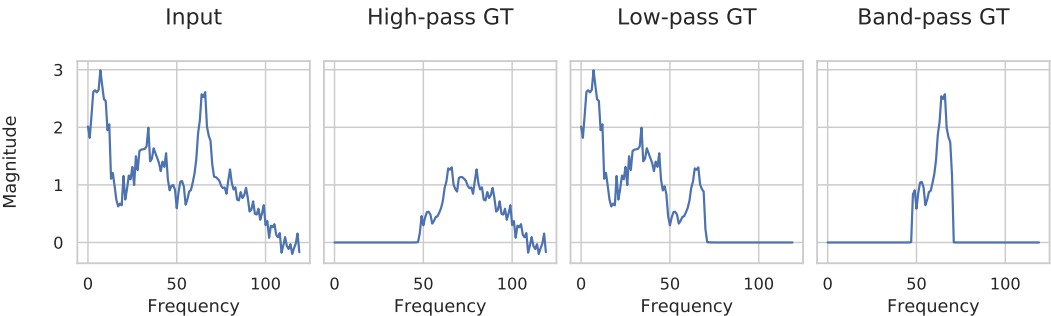

Figure 4: An example of graph spectrum in our SGS dataset and its corresponding high-pass, low-pass and band-pass filtering response using our hand-crafted filters.

# F MORE VISUALIZATIONS OF SPECTRUM

We compute spectrum as follows: suppose we have an undirected and unweighted graph with normalized adjacency matrix $A$ and node signal $X \in \mathbb{R}^{N \times D}$, where $N$ is the number of nodes and $D$ is the number of signal channels. Since $A$ is symmetric, we perform an eigen-decomposition on the adjacency matrix $A = U\Lambda U^T$. Then the spectrum of $X$ is computed by $S = U^T X$. More information about the graph spectrum and graph Fourier transformation can be found in Ortega et al. (2018).

Figure 5 shows the output spectrum of Vanilla GCN, 1st-Order GCN and SoGCN on the synthetic Band-Pass dataset. The visualizations are consistent to the Table 1 and Figure 3. Vanilla GCN almost loses all the band-pass frequency, resulting in a very poor performance. 1st-Order GCN learns to pass a part of medium-frequency band but still have an obvious distance from the groundtruth filter. SoGCN's filtering response is close to the groundtruth response, showing its strong ability to represent graph signal filters.

Figure 6 gives more spectrum visualizations on the ZINC dataset. We can observe the spectrum impacts of GRU on Vanilla GCN and our SoGCN. Each curve in the visualization figure represents the spectrum of each output channel, i.e., we plot each column of $S$ as a curve.

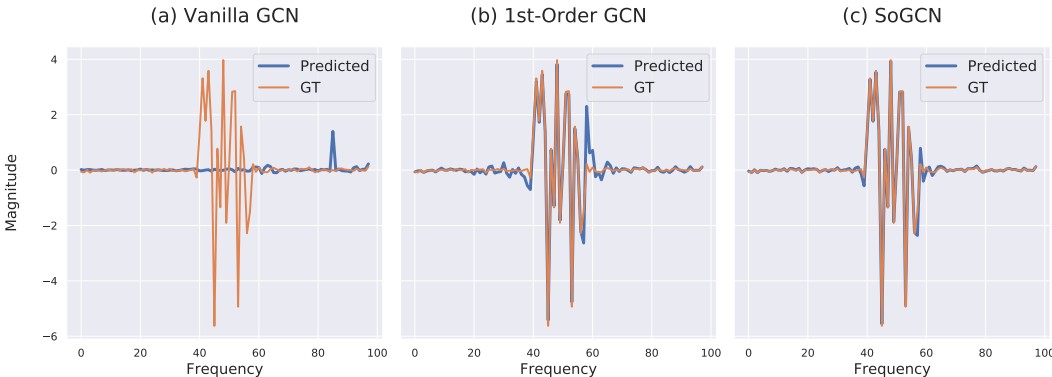

Figure 5: Visualizations of output spectrum on the SGS Band-Pass dataset.

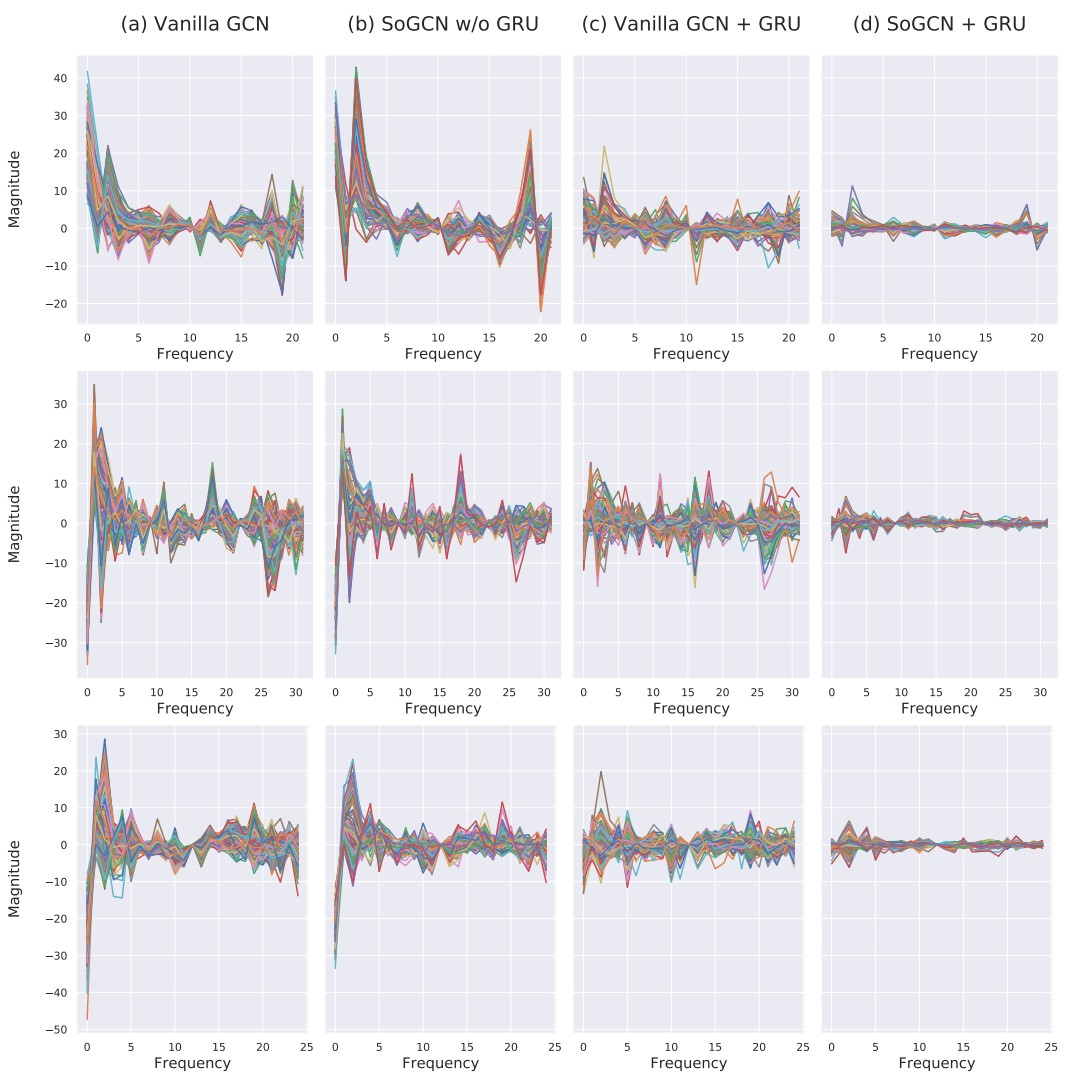

Figure 6: More visualizations of output spectrum on the ZINC dataset.

