# OpenReview forum: "SoGCN: Second-Order Graph Convolutional Networks"
_ICLR.cc/2021/Conference — Reject_

### Official Review · AnonReviewer4 · 2020-10-27
**Review for SoGCN: Second-Order Graph Convolutional Networks**

**Rating:** 5
**Confidence:** 4

**Review:**

This paper proposes a so-called second-order graph convolution, where an additional second-order term is introduced into traditional first-order graph convolution. The authors explain the merits of second-order graph convolution from the perspective of representation ability. The resulted second-order graph convolutional networks are compared with several graph convolutional networks on three benchmarks.

Strengths:
+: Incorporation of second-order information into graph convolution seems interesting, and such analogous idea has been verified in 2D convolutions.

+: The proposed method is simple and easy to implement.

Weaknesses:
-: The idea on incorporation of second-order or higher-order information into convolution is not novel. For example, Factorized Bilinear (FB) [r1] and Second-Order Response Transform (SORT) [r2] introduce second-order terms into traditional 2D convolutions, and they also claim second-order terms have better representation ability. Besides, second-order or higher-order information have also been used for global pooling for convolution networks [r3, r4, r5], which also show better representation ability. This paper lacks discussions on above these works, which will bring a side effect on contributions of this paper.
[r1] Factorized bilinear models for image recognition, ICCV 2017.
[r2] SORT: Second-order response transform for visual recognition, ICCV 2017.
[r3] Second-Order Pooling for Graph Neural Networks, arXiv, 2020.
[r4] Deep CNNs Meet Global Covariance Pooling: Better Representation and Generalization, TPAMI 2020.
[r5] Kernel pooling for convolutional neural networks, CVPR, 2020.

-: The experimental results are not very convincing.
(1) As shown in Table 2, pure soGCN achieves no improvement over compared methods, i.e., GatedGCN. GRU brings further gains for soGCN, but could GRU bring improvement for other compared methods?
(2) Number of parameters hardly represents model complexity totally. The model with the same number of parameters can have different computational complexity. Therefore, more metrics on model complexity (e.g., FLOPs) are suggested for comparison in Table 2.
(3) CIFAR10 and MNIST are too small and old to verify the effectiveness of different methods. The authors would better conduct experiments on more graph benchmarks. Besides, why higher-order  GCN are not compared on real-world benchmarks.
(4) Why parameters number of 3WLGNN is 100K on ZINC ?

-: The writing needs significant improvement.
(1) The authors would better give more detailed descriptions on differences between the proposed soGCN and related works ([Defferrard, 2016] and [Kipf&Welling, 2017]), further clarifying the contributions of the proposed method.
(2) I wonder the detailed computation methods and each curve in Fig.2, and why SoGCN is better than Vanilla GCN?
(3) The comparisons in terms of representation ability in section 4.3 is not very clear. The authors would better add a table to summarize representation ability of different graph convolution.
(4) Which method does MoNet indicate in Table 2 ?

---

> ### Author Response · Authors · 2020-11-21
> **Response to Reviewer4 - Part1**
>
> Thanks for your comments. To clarify our idea and make experiments more convincing, we provide the following responses.
>
> **1. All aforementioned works seem tangential to this work.**
>
> [r1]  introduced a Factorized Bilinear (FB) layer to model the pairwise feature interactions by considering the quadratic terms in the transformations.
>
> [r2] proposed a Second-Order Response Transform, which appends the element-wise product transform to the linear sum of a two-branch network module.
>
> [r3] proposed the second-order pooling as graph pooling (readout in our essay). It basically computes the inner-products between every node pair.
>
> [r4] proposed a global Matrix Power Normalized COVariance (MPN-COV) Pooling.
>
> [r5] proposed a general pooling framework that captures higher-order interactions of features in the form of kernels.
>
> All these works intend to compute higher-order feature interaction. But please note that, despite using the same term "order" with us, the meaning is totally different. In our work, we refer to "order" as the degree of the polynomial filters, which describes the node interaction w.r.t. the graph topology. In their works, the "order" means the order of channel interactions. Hence, our incorporation of second-order interaction is unrelated to those works mentioned above.
>
> **2. We supplemented the following explanation and experiments.**
>
> 1). First, GatedGCN is message-passing GNNs, which is orthogonal to our SoGCN. Our major improvements are made for vanilla GCN (spectral GCN). Without GRU, SoGCN can achieve at least comparable performance to GatedGCN. Second, applying GRU is to complement the low-pass filtering drawbacks of ReLU. This insight is more relevant to spectral GCNs. To refine our experiments, we add this ablation study for GRU on vanilla GCN, first-order GCN, third-order GCN as below.
>
> | Model             | ZINC (Test MAE) | MNIST (Test ACC) | CIFAR10 (Test ACC) |
> | ----------------- | --------------- | ---------------- | ------------------ |
> | Vanilla GCN       | 0.367$\pm$0.011 | 90.705$\pm$0.218 | 55.710$\pm$0.381   |
> | Vanilla GCN-GRU   | 0.295$\pm$0.005 | 96.020$\pm$0.090 | 61.332$\pm$0.849   |
> | SoGCN             | 0.238$\pm$0.017 | 96.785$\pm$0.113 | 66.338$\pm$0.155   |
> | SoGCN-GRU         | 0.201$\pm$0.006 | 97.729$\pm$0.159 | 68.208$\pm$0.271   |
> | 1st-order GCN     | 0.253$\pm$0.012 | 96.407$\pm$0.089 | 64.993$\pm$0.092   |
> | 1st-order GCN-GRU | 0.226$\pm$0.015 | 96.945$\pm$0.093 | 62.372$\pm$0.522   |
> | 3rd-order GCN     | 0.242$\pm$0.005 | 96.367$\pm$0.227 | 64.267$\pm$0.182   |
> | 3rd-order GCN-GRU | 0.203$\pm$0.001 | 97.375$\pm$0.052 | 64.242$\pm$0.511   |
> | 4th-order GCN     | 0.243$\pm$0.009 | 96.167$\pm$0.198 | 64.230$\pm$0.212   |
> | 4th-order GCN-GRU | 0.204$\pm$0.004 | 97.304$\pm$0.296 | 64.697$\pm$0.341   |
>
> Applying GRU to other message-passing models is off our context since they work under different mechanisms. One quick examination for GatedGCN + GRU on CIFAR10 is $67.295 \pm 0.555$, which indicates no performance gain compared with the original result.
>
> 2).  We are not using the parameter number to represent model complexity. Actually, the parameter number is used to constrain the size of models and range of hyper-parameters (e.g., width) for the sake of fair comparison. And our comparison criteria should strictly abide by the instruction of the benchmark [r1].
>
> [r1] Benchmarking Graph Neural Networks, arXiv 2020.
>
> 3). Indeed, for CNNs, CIFAR10 and MNIST are small and old, but they are still challenging for GCN/GNNs. Besides, our task is superpixel graph classification, which further poses a challenge to GCN/GNNs. Many GCN/GNNs have conducted experiments on this task. So it is reasonable to compare SoGCN with other models on these two superpixels datasets. To make our results more convincing, we added another two experiments on node classification datasets (PATTERN, CLUSTER) as below.
>
> | Models      | CLUSTER            | PATTERN            |
> | ----------- | ------------------ | ------------------ |
> | Vanilla GCN | $53.445\pm2.029$   | $63.880\pm0.074$   |
> | GAT         | $57.732\pm0.323$   | $75.824\pm1.823$   |
> | MoNet       | $58.064\pm0.131$   | $85.482\pm0.037$   |
> | GraphSage   | $50.454\pm0.145$   | $50.516\pm0.001$   |
> | GIN         | $58.384\pm0.236$   | $85.590\pm0.011$   |
> | GatedGCN    | $60.404\pm0.419$   | $84.480\pm0.122$   |
> | 3WLGNN      | $57.130\pm6.539$   | $85.661\pm0.353$   |
> | SoGCN       | $68.167\pm 1.164$  | $85.735 \pm 0.037$ |
> | SoGCN-GRU   | $67.994 \pm 2.619$ | $85.711\pm0.047$   |
>
> The state-of-the-art performance indicates our model's effectiveness. We have also presented ablation experiments for high-order GCNs in the previous paragraph for your reference.
>
> 4). We retrained 3WLGNN with 500K on ZINC. The test MAE was increased to 0.427$\pm$0.011.

---

> ### Author Response · Authors · 2020-11-21
> **Response to Reviewer4 - Part2**
>
>
> **3. We are sorry for confusing you with the writing problems. We will try to perfect the essay in the revision. Please let me clarify this issue as below.**
>
> 1). [Defferrard, 2016] formulated the polynomial graph filters (as we introduced in Section 2); vanilla GCN [Kipf\&Welling,2017] simplified a one-hop GC layer to be networks' basic building blocks. At first glance, SoGCN seems like a special case of [Defferrard, 2016], but our theory reveals that SoGC is the **smallest** filter with full expressiveness. This means one-hop GCs (e.g., vanilla GCN) are not enough; multi-hop GCs (degree larger than 2) are unnecessary. In this sense, SoGC stands out from other polynomial graph filters.
>
> 2). Each curve in the figure represents the spectrum of each output channel. We compute spectrum as follows: For any undirected graph with normalized adjacency matrix $A$, we perform an eigendecomposition on the adjacency matrix $A=X\Lambda X^{T}$. Then the spectrum of signal $S\in R^{N\times D}$ is computed by $\tilde{S}=X^{T}S$, where $N$ is the number of nodes and $D$ is the number of channels. We plot each column of $\tilde{S}$ as a curve in Figure 2. More information about the graph spectrum and graph Fourier transformation can be found in [r1].
>
> [r1] Graph Signal Processing: Overview, Challenges and Applications, IEEE 2017.
>
> 3). Thanks for your suggestions on a comparison table. We will consider adding one including more comparison in the revision. In terms of representational power, we provide the following "formula" for simplicity.
> $$
> \text{vanilla GCN} < \text{first-order GCN} < \text{second-order GCN} = \text{third-order GCN} = \cdots
> $$
>
> 4). Thanks very much for this suggestion. MoNet refers to the method proposed in *"Monti F, Boscaini D, Masci J, et al. Geometric deep learning on graphs and manifolds using mixture model CNNs. In Proceedings of the IEEE Conference on Computer Vision and Pattern Recognition. 2017: 5115-5124."*. We will cite it in our revision version.

---

### Official Review · AnonReviewer1 · 2020-10-27
**This is yet another paper on graph convolutional networks, considering a special case of high-order kernels.**

**Rating:** 5
**Confidence:** 3

**Review:**

This is yet another paper on graph convolutional networks (GCNs). The investigated SoGCN is a second-order GCN, thus a special case of high-order GCNs (namely with multi-hop graph kernels), which have has been proposed earlier by many researchers, such as by Defferrard et al. (2016), by Kipf & Welling (2017) and by Abu-El-Haija et al. (2019).

The main interest is that a second-order GC is a universal approximator, because any univariate polynomial can be factorized into sub-polynomials of degree two, which is not the case of first-order GCs.

The paper has some theoretical derivations that seem interesting. However, the proposed second-order GC is a special case of well-studied frameworks, which have been widely studied in the literature, such as with the expressive power analysis. As a consequence, the derived theoretical results seem less relevant.

In conducted experiments, the authors do not consider the same comparative analysis in all experiments. For example, two versions of the nonlinear activation are considered in experiments on CIFAR10 (with and without GRU); however, they were not considered in other experiments. Moreover, it is not clear where the combined RELU and GRU is considered.

---

> ### Author Response · Authors · 2020-11-20
> **Response to Reviewer1**
>
> We sincerely appreciate your comments on our work. However, we hope Reviewer 1 could re-examine our paper's contribution and experiments in the following aspects.
>
> **1. Our SoGCN is not a trivial case of higher-order GCNs.**
>
> Defferrard et al. (2016) introduced polynomial graph filters, Kipf & Welling (2017) then simplified to a one-hop GCN; Abu-El-Haija et al. (2019) proposed to adopt longer-hop propagation to enhance expressiveness. These works only concluded that one-hop GCN is not enough; multi-hop GCNs are beneficial. That is, these works only pointed out that the smallest propagation distance should be larger than 1. However, will keeping increasing the aggregation scale consistently benefit GCNs? How to set this propagation distance for each layer remains an open question, which leaves an extra hyper-parameter for high-order GCNs. Our work answers this question: it is sufficient to propagate two hops in each layer; other multi-hop GCNs are unnecessary in terms of improving performance. In this sense, SoGC is not a trivial case of multi-hop graph kernels. It is the smallest kernel that possesses the full representation power.
>
> **2. The method to analyze the powerfulness of GCNs is novel.**
>
> While a single polynomial GC layer has been well-studied in [r1] [r2], studying GCNs as a whole lacks considerable discussions. In [r2], the author intuitively claimed that it could recover a rich class of convolutional filter functions by stacking multiple vanilla GC layers. However, [r3] [r4] [r5] mathematically disproved this argument of vanilla GCN. But their theories can hardly provide constructive suggestions in finding a better graph kernel. There is no rigorous framework to distinguish the expressive power among higher-order GCNs yet.
>
> Instead, our proposed framework can evaluate the powerfulness of any spectral GCNs. Under this framework, we can not only reveal the drawbacks of vanilla GCNs similar to [r3] [r4] [r5], but also find it is sufficient to stack SoGC layers to achieve any convolutional filter by a well-known polynomial factorization theorem. Thus, SoGCN can be regarded as a straightforward corollary of our framework. In this sense, our derivation is not irrelevant at all.
>
> **3. With and without GRU have been considered in all experiments.**
>
> We added a comprehensive ablation study of spectral GCNs with and without GRU for CIFAR10, MNIST, and ZINC in the following table.
>
> | Model             | ZINC (Test MAE) | MNIST (Test ACC) | CIFAR10 (Test ACC) |
> | ----------------- | --------------- | ---------------- | ------------------ |
> | Vanilla GCN       | 0.367$\pm$0.011 | 90.705$\pm$0.218 | 55.710$\pm$0.381   |
> | Vanilla GCN-GRU   | 0.295$\pm$0.005 | 96.020$\pm$0.090 | 61.332$\pm$0.849   |
> | SoGCN             | 0.238$\pm$0.017 | 96.785$\pm$0.113 | 66.338$\pm$0.155   |
> | SoGCN-GRU         | 0.201$\pm$0.006 | 97.729$\pm$0.159 | 68.208$\pm$0.271   |
> | 1st-order GCN     | 0.253$\pm$0.012 | 96.407$\pm$0.089 | 64.993$\pm$0.092   |
> | 1st-order GCN-GRU | 0.226$\pm$0.015 | 96.945$\pm$0.093 | 62.372$\pm$0.522   |
> | 3rd-order GCN     | 0.242$\pm$0.005 | 96.367$\pm$0.227 | 64.267$\pm$0.182   |
> | 3rd-order GCN-GRU | 0.203$\pm$0.001 | 97.375$\pm$0.052 | 64.242$\pm$0.511   |
> | 4th-order GCN     | 0.243$\pm$0.009 | 96.167$\pm$0.198 | 64.230$\pm$0.212   |
> | 4th-order GCN-GRU | 0.204$\pm$0.004 | 97.304$\pm$0.296 | 64.697$\pm$0.341   |
>
> **4. We append a GRU after each ReLU to retain information loss.**
>
> We insert ReLU to increase the nonlinearity following the deep learning fashion. However, [r6] suggests that a contraction map, like ReLU, will trouble propagating information across a long range in a graph; [r4] [r5] suggested ReLU is a low-pass filter. Inspired by [r6] [r7], we append a GRU module to overcome these problems. GRU takes the signal before and after ReLU as two inputs. All SoGC layers share one GRU. We hypothesize that a GRU can be trained to suppress redundant spectral information, especially low-frequency information, and retain the lost features on the high-frequency band. We use visualization in Figure 5 (Appendix F) to support our conjecture. GRU's underlying mechanism is beyond the scope of our discussion.
>
> [r1] Convolutional Neural Networks on Graphs with Fast Localized Spectral Filtering, NIPS 2016.
>
> [r2] Semi-supervised Classification with Graph Convolutional Networks, ICLR 2017.
>
> [r3] Revisiting Graph Neural Networks: All We Have is Low-Pass Filters, arXiv 2019.
>
> [r4] Graph Neural Networks Exponentially Lose Expressiveness for Node Classification, ICLR 2020.
>
> [r5] A Note on Over-Smoothing for Graph Neural Networks, ICML 2020.
>
> [r6] Gated Graph Sequence Neural Networks, ICLR 2016.
>
> [r7] Neural message passing for quantum chemistry, ICML 2017.

---

### Official Review · AnonReviewer3 · 2020-10-28
**Second-Order Graph Convolutional Networks**

**Rating:** 5
**Confidence:** 5

**Review:**

The submission identifies the importance of second-order filter by showing that two is the minimally necessary order to achieve full representation power. Based on this observation, this paper proposes Second-Order Graph Convolutional Networks. The proposed method is evaluated on graph classification benchmarks and demonstrates good empirical performance. As far as I know, the proposed method is novel and can inspire future analysis on the expressive power of graph convolution.

Strength:
- The proposed method, SoGC, is theoretically motivated by the fact that second order graph convolution has universal representation power (Thm. 1)
- The proposed method demonstrates strong empirical performance on synthetic and real-world datasets

Weakness and Questions:
- The usage of GRU feels ad-hoc to me. Without using GRU, the proposed method in fact does not achieve state-of-the-art performance on CIFAR10 and MNIST, which is concerning. I understand that the authors attempt to analyze the effect of GRU in Section 5.1. However, the claim that "GRU retains information from previous layers effectively ..." needs more supporting evidence than Figure 2, which is conducted on a single graph from a single dataset. I encourage the authors to provide more in-depth analysis of GRU and connects it to the rest of the paper.
For more low-level questions:
     - How is the input node order to the GRU decided? Will changing the input order affect results?
     - In Figure 2, is the number of parameters controlled?
     - Would GRU also improves the performance of vanilla GCN? As Figure 2 showed, GRU also changes the signal of a vanilla GCN significantly.
- I would like to see an ablation study between the non-linear and linear versions of SoGCN. Given that in section 3 the analysis is only done for the linear version, I am curious about how well the linear model can perform. Also, the theory predicts that the linear Second Order GCN should outperform linear Vanilla GCN (this is like SGC?). I would like to see this prediction gets tested.
- As the authors acknowledge in the introduction, Abu-El-Haija et al. (2019), MixHop, makes similar observations about higher-order GCN. I am curious about how these two models compare to each other performance-wise.
- Minor note: The submission only evaluates the proposed method on graph classification task. Will the proposed method works for node classification or link prediction? Even if it doesn't, the authors should document the limitations of this method if it doesn't work for those tasks.

Typos:
- Page 1, the second to last paragraph: "in channel-wise filtering Furthermore" -> missing period
- Page 6, "we can whiteness the ineffectiveness" -> "witness"

---

> ### Author Response · Authors · 2020-11-21
> **Response to Reviewer3 - Part1**
>
> We greatly thank Reviewer3 for the thoughtful comments and for recognizing the novelty of our SoGCN models and analyzing methods.
>
> **1. For your concerns on GRU, please see our responses as follows.**
>
> Spectral GCNs are separated from message-passing GNNs. Even without GRU, our SoGCN can promote vanilla GCNs comparable to state-of-the-art results (vanilla GCNs is considered flawed).
>
> The critical point of this work is to use SoGC to achieve more diverse filters (span the whole filter space) and distinguish richer features on the spectrum. Our usage of GRU shares similar insights. On the real datasets, we insert ReLU to introduce nonlinearity following the deep learning fashion. However, [r1] [r2] pointed out ReLU is a low-pass filter, which causes a side effect on what we expect on SoGCNs (i.e., causing over-smoothing). GRU leverages gate mechanism to preserve and forget information [r3]. We hypothesize that a GRU can be trained to suppress redundant low-frequency information and retain the lost features on the high-frequency band. Inspired by [r4] [r5], we shared one GRU among each layer, which takes the signal before and after ReLU as two inputs. We provide more visualization in Figure 5 to support our conjecture. Discussing GRU's underlying mechanism is beyond the scope of our work, and remains an open question for future study.
>
> For your specific problems, our responses are as below.
>
> * We think Reviewer3 was confused by our usage of GRU. The inputs of GRU are node sequences. Our usage of GRU resembles GGNN [r4], MPNN [r5], instead of LSTM in GraphSage [r6]. That is, GRU takes the signal before and after ReLU as two inputs (more like a temporal sequence). Therefore, input node order makes no impact. To avoid misunderstanding by other readers, we will improve the writing of this part carefully in the revision.
>
> * Parameter number is controlled. The illustration in Figure 2 and Figure 5 directly comes from the experiments on ZINC shown in Table 2.
>
> * As we discussed above, GRU can relieve the over-smoothing problem of spectral GCNs. We defer the results of Vanilla GCN + GRU to the next paragraph. Vanilla GCN + GRU cannot outperform our models due to the deficient power of its graph filtering part.
>
> [r1] Graph Neural Networks Exponentially Lose Expressiveness for Node Classification, ICLR 2020.
>
> [r2] A Note on Over-Smoothing for Graph Neural Networks, ICML 2020.
>
> [r3] Learning Phrase Representations using RNN Encoder–Decoder for Statistical Machine Translation, arXiv 2014.
>
> [r4] Gated Graph Sequence Neural Networks, ICLR 2016.
>
> [r5] Neural message passing for quantum chemistry, ICML 2017.
>
> [r6] Inductive Representation Learning on Large Graphs, NIPS 2017.
>
> **2. For your concerns on the linear model, our response is as below.**
>
> Indeed, SGC exhibited comparable performance to vanilla GCN with nonlinearity on citation network datasets. However, it is not the case for other large-scale datasets. To verify this argument, we conduct a quick experiment as below on the ZINC dataset. The experiment results indicate a performance gain over vanilla GCNs by 0.42.
>
> | SGC  | Linear SoGCN |
> | ------ | ------------------- |
> |0.650  | 0.608 ($\downarrow 0.42$) |
>
> It is noteworthy that, the synthetic experiments (Table 1) are designed to verify our theories for linear GCNs. They do not incorporate nonlinearity.

---

> ### Author Response · Authors · 2020-11-21
> **Response to Reviewer3 - Part2**
>
>
> **3.  We supplement an additional ablation study for higher-order models.**
>
> First, MixHop Abu-El-Haija et al. (2019) repeats mixing feature representations of neighbors at various hops to capture richer features. But our paper basically doubts if larger hops can keep bringing benefits. In terms of filter representation power, our theory tells that one-hop is not enough, while two-hop is sufficient. MixHop only pointed out the propagation distance should be larger than 1, while we added that exact two-hop propagation achieves the maximal filtering power. Second, we note that our theoretical framework differs from MixHop. By their means, it is hard to reach our conclusion. The supplemented table below shows higher-order (3th- or 4th-order) have no improvement than SoGCN.
>
> | Model             | ZINC (Test MAE) | MNIST (Test ACC) | CIFAR10 (Test ACC) |
> | ----------------- | --------------- | ---------------- | ------------------ |
> | Vanilla GCN       | 0.367$\pm$0.011 | 90.705$\pm$0.218 | 55.710$\pm$0.381   |
> | Vanilla GCN-GRU   | 0.295$\pm$0.005 | 96.020$\pm$0.090 | 61.332$\pm$0.849   |
> | SoGCN             | 0.238$\pm$0.017 | 96.785$\pm$0.113 | 66.338$\pm$0.155   |
> | SoGCN-GRU         | 0.201$\pm$0.006 | 97.729$\pm$0.159 | 68.208$\pm$0.271   |
> | 1st-order GCN     | 0.253$\pm$0.012 | 96.407$\pm$0.089 | 64.993$\pm$0.092   |
> | 1st-order GCN-GRU | 0.226$\pm$0.015 | 96.945$\pm$0.093 | 62.372$\pm$0.522   |
> | 3rd-order GCN     | 0.242$\pm$0.005 | 96.367$\pm$0.227 | 64.267$\pm$0.182   |
> | 3rd-order GCN-GRU | 0.203$\pm$0.001 | 97.375$\pm$0.052 | 64.242$\pm$0.511   |
> | 4th-order GCN     | 0.243$\pm$0.009 | 96.167$\pm$0.198 | 64.230$\pm$0.212   |
> | 4th-order GCN-GRU | 0.204$\pm$0.004 | 97.304$\pm$0.296 | 64.697$\pm$0.341   |
>
> MixHop does not provide code on the benchmarks. So we are not able to compare MixHop with ours fairly. Theoretically, the performance of MixHop and our higher-order models should be approaching.
>
> **4. Our model achieves state-of-the-art results on node classification**
>
> Thanks for your suggestions. We evaluate our SoGCN on CLUSTER and PATTERN datasets for node classification tasks and demonstrate the state-of-the-art results. The results are presented as below. However, as we only focus on node-domain feature extraction, SoGCN does not incorporate edge representations. SoGCN may not be suitable for link prediction, as suggested by [r1].
>
> | Models      | CLUSTER            | PATTERN            |
> | ----------- | ------------------ | ------------------ |
> | Vanilla GCN | $53.445\pm2.029$   | $63.880\pm0.074$   |
> | GAT         | $57.732\pm0.323$   | $75.824\pm1.823$   |
> | MoNet       | $58.064\pm0.131$   | $85.482\pm0.037$   |
> | GraphSage   | $50.454\pm0.145$   | $50.516\pm0.001$   |
> | GIN         | $58.384\pm0.236$   | $85.590\pm0.011$   |
> | GatedGCN    | $60.404\pm0.419$   | $84.480\pm0.122$   |
> | 3WLGNN      | $57.130\pm6.539$   | $85.661\pm0.353$   |
> | SoGCN       | $68.167\pm 1.164$  | $85.735 \pm 0.037$ |
> | SoGCN-GRU   | $67.994 \pm 2.619$ | $85.711\pm0.047$   |
>
> [r1] Benchmarking Graph Neural Networks, arXiv 2020.
>
> **5. Thanks so much for your correction. We will fix these typos in the revision.**

---

### Official Review · AnonReviewer2 · 2020-10-29
**Simple method, interesting observations**

**Rating:** 7
**Confidence:** 4

**Review:**

The authors argue that second-order graph convolutions (SoGC) should be the building blocks for future graph networks. The argument is that some of the second-order functions cannot be represented by stacking first-order graph convolutions. In contrast, second-order can represent any higher-order ones. In general, the theory is sound and the results on the synthetic data and ZINC are strong and I don't see many flaws, but I would like to see more results on realistic datasets.

##### Strengths
- The method is quite simple.
- The results on the synthetic dataset and ZINC are quite impressive.
- The visualization in Figure 5 convinces me that SoGC can deal with the oversmoothing problem of vanilla GCNs.
- The theoretical insights and proofs look correct to me, while I am not 100% confident.

##### Weaknesses
- MNIST and CIFAR 10 do not seem to be real use cases for Graph models. According to the [homepage of MNIST](http://yann.lecun.com/exdb/mnist/), none of the MNIST results in Table 2 is better than an SVM with degree 4 polynomial kernels, a 2-layer MLP with 800d, or a LeNet-1 which has only 3k parameters instead of 100k. It would be better to see results on more realistic datasets where graph networks are close to state-of-the-art.

---

> ### Author Response · Authors · 2020-11-20
> **Response to Reviewer2**
>
> Thanks for your positive feedback and thanks for recognizing our work's strengths in theories, simplicity, visualization, and experiments. For your concerns on the selection of datasets, our response is as below:
>
> Indeed, the performance of all GCN/GNNs indicates their disadvantages on MNIST and CIFAR10 datasets, compared with CNN, SVM, and MLP. The performance bottleneck of GCN/GNNs poses new challenges for GCN/GNNs. Thus, the latest GNN benchmark [r1] includes the two datasets as sanity-checks, as we expect most GNNs to perform close to 100\% for MNIST and well enough for CIFAR10.
>
> In addition, the classical image classification task on MNIST and CIFAR 10 turns to "superpixel" graph classification here, which is incompatible with CNNs but potentially a good application for GCN/GNNs. Images are converted into graphs by the superpixel algorithm [r2], which assigns each node’s features as the super-pixel coordinates and intensity. The SoGCN achieves state-of-the-art performances among GCN/GNN models on several datasets, showing SoGCN's effectiveness.
>
> Moreover, we supplement two experiments on CLUSTER and PATTERN datasets for node classification, where our method again outperforms other GCN/GNN models. This further testifies the effectiveness of our SoGCN.
>
> | Models      | CLUSTER (Test ACC)           |  PATTERN (Test ACC)            |
> | ----------- | ------------------ | ------------------ |
> | Vanilla GCN | $53.445\pm2.029$   | $63.880\pm0.074$   |
> | GAT         | $57.732\pm0.323$   | $75.824\pm1.823$   |
> | MoNet       | $58.064\pm0.131$   | $85.482\pm0.037 $   |
> | GraphSage   | $50.454\pm0.145$   | $50.516\pm0.001$   |
> | GIN         | $58.384\pm0.236$   | $85.590\pm0.011$   |
> | GatedGCN    | $60.404\pm0.419$   | $84.480\pm0.122$   |
> | 3WLGNN      | $57.130\pm6.539$   | $85.661\pm0.353$   |
> | SoGCN       | $68.167\pm 1.164$  | $85.735 \pm 0.037$ |
> | SoGCN-GRU   | $67.994 \pm 2.619$ | $85.711\pm0.047$   |
>
> ---
>
> **References:**
>
> [r1] Benchmarking Graph Neural Networks, arXiv 2020.
>
> [r2] SLIC superpixels compared to state-of-the-art superpixel methods, TPAMI 2012.

---

> > ### Comment · AnonReviewer2 · 2020-11-20
> > **Response to author's feedback**
> >
> > Thank you for providing additional results on these datasets, which makes it more convincing. Additionally, I'm glad to see an ablation experiment on GRU, and different orders of graph convolution are provided in the response to R1, which is also what I'm looking for. (I appreciate R1 for pointing this out.) While some other reviews may have different opinions, I think the theoretical contribution of this paper is interesting and the experimental results are strong.
> > Admittedly, there are valid concerns about related works from other reviewers. I hope the authors can fix them in a revised version of the paper during the discussion period.
> >
> > Overall, I still keep my original judgment and vote for accepting this paper. As long as the authors can update the paper to address some writing issues brought by other reviewers, I don't see any reason why we should reject it.

---

### Author Response · Authors · 2020-11-24
**Revision Summary**

We have updated our revision with respect to reviewers' suggestions. The major revised parts are as follows:

1. Section 5.2: The comparison with related works to clarify our novelty and contribution.
2. Section 5.1: The introduction to GRU modules. We explain how GRU is integrated and benefits spectral GCNs with empirical study.
3. Section 6.2: We add an ablation study on different-order GCNs with and without GRU.
4. Section 6.2: We supplement our SoGCN's performances on two additional datasets (CLUSTER and PATTERN) for node classification.

---

### Decision · Program_Chairs · 2021-01-07
**Final Decision**

**Decision:**

Reject

**Comment:**

Four reviewers have reviewed this paper. After rebuttal, the reviewers' recommendations were borderline. Rev. 4 remains concerned about relation of second-order approaches in CV and second-order filters. Indeed, there exists a connection although it is perhaps subtle in its nature and equally concerning is the connection with general Polynomial filters in many GCN papers. As other reviewers point out, MixHop and Jumping Knowledge also allow multi-hop designs. More importantly, APPNP and SGC networks allow multiple hops. From that point of view, the proposed approach is rather a recap of existing observations and contributions. Finally, even Rev. 2 has indicated that the paper is perhaps 'average' after checking with comments of other reviewers. Therefore, at this point, the paper is slightly below the acceptance threshold.